# OpenReview forum: "Geometri-Disentangelment Unlearning"
_ICLR.cc/2026/Conference — ICLR 2026 Conference Desk Rejected Submission_

### Official Review · Reviewer_LTCV · 2025-10-23

**Soundness:** 3
**Presentation:** 4
**Contribution:** 4
**Rating:** 6
**Confidence:** 3

**Summary:**

With the goal of unlearning a target model while preserving its performance on the remaining data, this paper first provides conditions for the retain set loss to remain unchanged. Then it proposes an unlearning method that decomposes the forgetting signal into tangential and normal components to the retain space, preserving only the normal one. Specifically, the authors demonstrate that, to first order, the retain set loss remains unchanged when the weight update is orthogonal to the subspace spanned by the retain gradients. Therefore, the authors propose geometric-disentanglement unlearning (GU), which projects forget gradients to the normal component of the subspace spanned by retain gradients that guarantee the steepest descent, under a trust-region budget. Results in established benchmarks for LLMs (TOFU, MUSE, WMDP) demonstrate the effectiveness of the proposed method in mitigating the retain set performance degradation.

**Strengths:**

1. This paper provides the first theoretically grounded representation of the entanglement between retain and forget datasets, showing, under minimal assumptions, which update direction jointly leaves the retain loss locally invariant while unlearning the forget set.
2. Rigorous theoretical proofs support every claim in the paper.
3. I appreciate that the authors submitted the code used in their experiments, allowing full reproducibility.
4. Although some steps are somewhat abstracted, the writing is clear, and the notation is consistent and appropriate.

**Weaknesses:**

**Major weaknesses**
1. **Comparison with similar approaches.** Although I believe the results section is extensive, showing that the proposed method, when paired with existing unlearning algorithms, improves the unlearning/retaining trade-off, comparing this method with other approaches that propose orthogonal projections [1,2](this list is non-exhaustive) would have strengthened the paper's claims.
2. **The extent to which this method works.** It is not clear to me to what extent this method works because of the following two concerns:
    1. First, it is true that in most cases this approach benefits unlearning; yet, retain set improvements are almost always mild and clearly depend on the effectiveness of the underlying unlearning algorithm. For instance, taking Table 1, under a 10% unlearning ratio and Llama 1B, GradDiff scores 0.12 ES Re. from the original retaining performance of ~0.65. By applying the proposed GU, it reaches 0.15. Similarly, with DPO, the score passes from 0.26 to 0.28. There are a few cases where the improvement is stronger, like in UNIDIAL (from 0.26 to 0.69) and SatImp.
    2. I am unsure about the effectiveness of the proposed method under highly entangled forget and retain spaces. The forget and retain identities in TOFU are probably well disentangled; therefore, the proposed method is shown to be somewhat effective. Instead, the GU performance improvements are more subtle and sometimes questionable in MUSE and WMDP. My intuition is that forget and retain datasets for these two benchmarks are much more entangled (e.g., MUSE forget set contains actual Harry Potter books, while the retain set contains knowledge crawled from open wikis). Thus, GU cannot properly find an unlearning direction orthogonal to the retain set subspace. Bringing this to the extreme case, I do not know whether this approach can work in the random unlearning scenario.
3. **Hyperparameter search.** Some results are a bit weird to me. For instance, CEU is shown to perform very poorly, but with the proposed GU it substantially improves its performance. Yet, the CEU paper reports results that are far better than those highlighted here, although with Llama 2 instead of Llama 3. As it is well known that hyperparameter selection is of paramount importance for machine unlearning to achieve satisfactory results [3,4,5], this makes me question the hyperparameter search and selection of this paper. So, is the proposed method actually improving the baselines, or does it make up for improper hyperparameter tuning? Additionally, the hyperparameter search and selection for this paper is not reported; likewise, a study on the hyperparameter sensitivity of GU.

**Minor weaknesses/suggestions**
1. L.57 states that the objective is to "reduce forgetting loss while leaving the retained knowledge unchanged [...]." This is partly correct for the forget loss, as, e.g., in gradient ascent, the loss increases.
2. L.106 states that for the forget set, the update applies gradient ascent. Yet, L.121-129 correctly state that this is one of the possible choices. I'd suggest rewriting the L.106 statement, taking this into account.
3. Although Appendix B reports full details about the algorithm implementation and the submission also comes with the code, I suggest adding a pseudocode in Appendix B to show the high-level general steps to execute GU; otherwise, it is a bit cryptic in my opinion.

[1] Sendera, Marcin, et al. "SEMU: Singular Value Decomposition for Efficient Machine Unlearning." arXiv, 2025.\
[2] Hoang, Tuan, et al. "Learn to unlearn for deep neural networks: Minimizing unlearning interference with gradient projection." WACV, 2024.\
[3] Cadet, Xavier F., et al. "Deep unlearn: Benchmarking machine unlearning." EuroS&P, 2024.\
[4] Kim, Hyoseo, Dongyoon Han, and Junsuk Choe. "Negmerge: Consensual weight negation for strong machine unlearning." ICML, 2025.\
[5] He, Zhengbao, et al. "Towards natural machine unlearning." TPAMI, 2025.

**Questions:**

1. Could the authors provide a comparison with at least one orthogonal projection method in the machine unlearning literature? Or an intuition for why the comparison was excluded in the first place.
2. Could the authors elaborate on the performance improvements of GU in relation to weaknesses 2.1 and 2.2 (W2.1, W2.2)?
3. Could the authors report the hyperparameters chosen and how they were searched for? Additionally, I would also like to see the hyperparameter sensitivity analysis for GU.

**Motivation for my score**\
Although some results are not convincing, I believe the community will benefit from the theoretical insights of this paper; therefore, I positively recommend this work, yet I am still reluctant to assign an accept score, which I'll reconsider after the authors' rebuttal.

---

> ### Author Response · Authors · 2025-11-21
>
> We sincerely thank the reviewer for the strong recognition of our work and for acknowledging our theoretical contributions, rigor, and reproducibility efforts. We appreciate that reviewer pzAp positively recommends our work. Below, we address reviewer pzAp’s constructive feedback regarding related works, the scope of effectiveness, and hyperparameter tuning.
>
> **Summary**
>
>
> We summarize our core response:
> 1.  **Pareto Dominance:** GU is designed to optimize the trade-off between forgetting and retention. Our new analysis shows that GU achieves **joint improvement** (better or unchanged retention/utility and forgetting) in the large majority of configurations (**79.2%**), effectively pushing the Pareto frontier of existing unlearning algorithms. We add a Pareto analysis Figure in our paper for your reference (Figure 2).
> 2.  **Other Baselines & Scalability:** Unlike prior projection methods (UNSC/PGU/SEMU) that rely on computationally prohibitive activation or Gram statistics (infeasible for LLMs), GU introduces a scalable geometric constraint, making it uniquely suitable for large LLMs.
> 3.  **Safety Under Entanglement:** In highly entangled settings (potentially MUSE/WMDP or even random unlearning), GU behaves as a conservative, retain-safe filter rather than an over-aggressive unlearning operator.
> 4. **Fairness in Tuning:** We use a shared, transparent hyperparameter protocol provided by open-unlearning for both baselines and GU-augmented methods to ensure fair comparison.
>
>
> **QW1. Comparison with Orthogonal-Projection / Null-Space Methods**
>
>
> We appreciate the references to SEMU [1], Hoang et al. [2], Cadet et al [3], Kim et al [4], and He et al [5]. While conceptually related, GU differs in **subspace definition**, **geometric metric**, and **scalability to LLMs**. Please kindly refer to the response 2 to reviewer 1. To be more specific:
>
> * **Subspace Definition:** Prior methods rely on heuristics such as feature statistics [2] or SVD on forget gradients [1] to construct "forget-related" subspaces. GU instead defines the subspace as the **retain tangent space** with theoretical soundness:
>     $$T_r(\theta) = \mathrm{span}\{\nabla_\theta \ell_r(x_r;\theta)\}.$$ Add a sentence here about why it’s good. For example, “it helps us identify the entanglement as the tangential part of unlearning and retaining gradients. We appreciate the reviewer’s recognition of our definition driven by theoretical insights instead of heuristics”.
> * **Geometry:** Existing projection-based methods use Euclidean metrics. GU uses the **optimizer-induced SPD metric** ($H$) (from Adam) and constructs an ($H$)-orthogonal projector. This aligns the projection with the actual update dynamics and local curvature in high-dimensional LLM parameter space.
> * **Scalability:** Methods like UNSC/PGU require storing and factorizing large covariance or Gram matrices, which is infeasible for billion-parameter LLMs. SEMU requires repeated SVD on weight/gradient matrices, which quickly becomes expensive at scale. GU instead maintains a lightweight streaming basis with rank ($k \ll d$) per parameter block, leading to $O(pk)$ overhead and making it practical for 1B–8B parameter models.
>
>
> We added (i) a comparison table in the related work section highlighting these differences, and (ii) include an ablation reference to the rebuttal for reviewer 2 that replaces our retain-gradient subspace with Ecilidian subspaces $I$, to empirically demonstrate the necessity of our retain-tangent, $H$-orthogonal construction.

---

> > ### Author Response · Authors · 2025-11-21
> >
> > (Continuing from the previous section)
> >
> > **WQ2. Scope of Effectiveness: "Mild" Gains & Entanglement**
> >
> >
> > **W2.1. "Mild" Improvements vs. Pareto Stability (W2.1)**
> >
> >
> > We agree that in some easier TOFU settings the absolute gain in $ES_{Re}$ can appear numerically small (e.g., 0.12 $\to$ 0.15). However, unlearning is inherently a multi-objective problem. The goal of GU is not to maximize a single scalar metric, but to act as a **geometric safety guardrail** that pushes the **Pareto frontier** of existing algorithms.
> >
> > * **Quantitative Robustness (Joint Improvement):** To address the concern about gains, we conducted a holistic analysis across all experimental configurations (Dataset $\times$ Model Size $\times$ Base Method):
> >     * GU improves or maintains **Retain Quality ($ES_{Re}$)** in **91.7%** of cases in Table 1.
> >     * GU improves or maintains **Model Utility (MU)** in **86.1%** of cases in Table 1.
> >     * Crucially, we observe **Joint Improvement** (where $ES_{Un}$ does not worsen and $ES_{Re}$/MU improve or stay neutral) in **79.2%** of cases.
> >     * This shows that GU consistently moves methods towards the ideal Pareto zone (high retention, high forgetting) relative to their vanilla counterparts, even when the absolute change on a single metric is small.
> > * **Hard Regimes:** In more difficult regimes—larger models and higher forget ratios—baselines often achieve strong forgetting only by severely harming the retain distribution (e.g., near-collapse of MU or $ES_{Re}$). In these cases, GU yields more visible gains: it prevents collapse and restores retain metrics while keeping $ES_{Un}$ low. We will make these "hard regime" cases (e.g., UNDIAL/CE-U on 3B, SimNPO on 8B) more prominent in the revised analysis.
> >
> >
> > **W2.2. Highly Entangled Settings (MUSE, WMDP, and Random Unlearning) (W2.2)**
> >
> >
> > We agree that MUSE and WMDP represent highly entangled scenarios. This is precisely where naive unlearning objectives tend to push too far:
> > * On these benchmarks, many baselines achieve strong forgetting only at the cost of severely degrading retain performance (large drops in MU, ROUGE, or general accuracy).
> > * GU, by construction, enforces a retain-aware geometric constraint, so it naturally produces a **more conservative but safer trade-off**: in our experiments, GU often keeps or improves retain metrics while still reducing $ES_{Un}$, instead of chasing the most aggressive forgetting.
> >
> >
> > From a theoretical standpoint, your intuition about entanglement is exactly right:
> > * If the forget gradient lies largely inside the retain tangent space ($T_r(\theta)$), then the retain-orthogonal component ($g_\perp$) is necessarily small. In that regime, any method that insists on first-order retain invariance cannot produce large unlearning steps without harming the retain set.
> > * GU is a **hard-constraint** method: when the available retain-orthogonal direction is small, it intentionally takes small steps rather than moving in directions that would damage $L_r$. We view this conservative behavior as desirable in safety-critical deployments.
> >
> >
> > This also explains the **random unlearning** extreme you raised. If $D_f$ and $D_r$ are random splits of the same distribution, forget and retain gradients become nearly collinear on average, and retain-orthogonal directions vanish. In this setting, strong unlearning and strict retain invariance are fundamentally incompatible. GU effectively detects this incompatibility and behaves conservatively, preventing the model from being destroyed in a futile attempt to unlearn. We argue this is a desirable safety feature: **when safe unlearning is mathematically impossible, GU prevents catastrophic damage.** We will explicitly acknowledge this limitation and add a small synthetic "random unlearning" experiment in the appendix to illustrate this behavior.
> >
> >
> > **Future Work:** We agree that in highly entangled benchmarks one might want a tunable trade-off. A natural extension of GU is to relax the hard orthogonality constraint into a **soft constraint**, e.g., penalizing (rather than removing) the retain-aligned component. This would allow practitioners to deliberately trade a controlled amount of retain degradation for stronger forgetting when appropriate. We will discuss this extension in the conclusion.

---

> > > ### Author Response · Authors · 2025-11-21
> > >
> > > (Continuing from the previous section)
> > >
> > >
> > > **WQ3. Hyperparameter Search & CE-U**
> > >
> > >
> > > We carefully designed our hyperparameter protocol to ensure that GU’s improvements are not artifacts of more favorable tuning.
> > >
> > >
> > > * **Shared Protocol:** We follow the OpenUnlearning framework [3]: for each unlearning objective, we tune over the same grid of learning rates and loss-weight hyperparameters. For the GU-augmented versions, we reuse exactly the same loss hyperparameters as the underlying method; GU only adds the projection on the forget gradient. Because the projection shrinks the feasible gradient component, the effective step size of GU is actually smaller than that of the baseline at the same learning rate. To compare at comparable "unlearning strength", we tune both baseline and baseline+GU on identical LR grids and select each method’s best configuration based on validation metrics. For the added hyperperameters, please kindly reference to the rebuttal for reviewer 2.
> > > * **GU Hyperparameters:** GU introduces a small number of additional hyperparameters (trust-region parameters $\kappa, \tau$ and basis size $k$), which we fix to a single global setting across all models, benchmarks, and forget ratios. In the revision, we will add a brief sensitivity study in the appendix showing that GU’s Pareto improvements are stable over a reasonable range of $\kappa$ and $k$.
> > > * **CE-U Specifics:** For CE-U in particular, we faithfully implement the original formulation [4]: the CE-U loss itself does not include any explicit retain term, positive samples, or oracle/reference model. This aligns with the CE-U paper's stated goal. In our LLaMA-3 setting, CE-U alone can be quite aggressive and sometimes destabilizes retain performance. Adding GU on top of CE-U acts as a **stabilizer**: CE-U + GU maintains strong forgetting while significantly mitigating utility collapse. We will clarify this in the revision and provide a hyperparameter table in Appendix D detailing search ranges and selected values.
> > >
> > >
> > > **R4. Minor Corrections**
> > >
> > >
> > > We (i) corrected the wording regarding loss directions in L.57 (now in L58) and L.106 to clearly distinguish gradient ascent on the forget loss from gradient descent on the retain loss, and (ii) add pseudocode in Appendix B to summarize the high-level GU training step following your suggestion.
> > >
> > >
> > > We believe these clarifications and planned revisions address your concerns about comparisons, effectiveness under entanglement, and hyperparameter fairness. We again thank you for your positive recommendation and hope to secure your full support.
> > >
> > >
> > > [1] Sendera, Marcin, et al. "SEMU: Singular Value Decomposition for Efficient Machine Unlearning." arXiv preprint arXiv:2502.07587 (2025).
> > >
> > > [2] Hoang, Tuan, et al. "Learn to unlearn for deep neural networks: Minimizing unlearning interference with gradient projection." Proceedings of the IEEE/CVF Winter Conference on Applications of Computer Vision. 2024.
> > >
> > > [3] Cadet, Xavier F., et al. "Deep unlearn: Benchmarking machine unlearning." EuroS&P, 2024.
> > >
> > > [4] Kim, Hyoseo, Dongyoon Han, and Junsuk Choe. "Negmerge: Consensual weight negation for strong machine unlearning." ICML, 2025.
> > >
> > > [5] He, Zhengbao, et al. "Towards natural machine unlearning." TPAMI, 2025.
> > >
> > > [6] Dorna, Vineeth, et al. "OpenUnlearning: Accelerating LLM Unlearning via Unified Benchmarking of Methods and Metrics." arXiv preprint arXiv:2506.12618 (2025).
> > >
> > > [7] Yang, Bo. "CE-U: Cross Entropy Unlearning." arXiv preprint arXiv:2503.01224 (2025).

---

> > > > ### Comment · Reviewer_LTCV · 2025-11-24
> > > >
> > > > I thank the authors for providing such detailed responses. I find the explanation about comparing with the Orthogonal-Projection and Null-Space methods convincing. Additionally, I liked the proposed limitations and future works to be added. However, I am still reluctant about the performance gains that GU brings:
> > > > - Could the authors provide the average improvement (per-metric and per-dataset) when using GU?

---

> > > > > ### Author Response · Authors · 2025-11-25
> > > > > **Response: Average improvement (per-metric and per-dataset) when using GU**
> > > > >
> > > > > We sincerely appreciate that you found our responses helpful, and we appreciate your recognition of the distinctions between our method GU and prior orthogonal-projection approaches. We value the opportunity to further demonstrate how GU performs relative to existing baselines.
> > > > >
> > > > > To make this explicit, we computed **per-dataset, per-metric average improvements** when augmenting each baseline with GU. For each metric $m$, we define a signed improvement
> > > > >
> > > > > $$
> > > > > \begin{align}
> > > > > \Delta_m :=
> > > > > m_{\text{GU}} - m_{\text{base}}, \text{if higher is better} \quad \quad \\
> > > > > m_{\text{base}} - m_{\text{GU}}, \text{if lower is better}
> > > > > \end{align}
> > > > > $$
> > > > >
> > > > >
> > > > > Thus, $\Delta_m > 0$ indicates an improvement on metric m, whether that corresponds to higher retention/utility or lower unlearning error.
> > > > >
> > > > > To make this effect more concrete, we added Pareto-frontier visualizations in the Experiment Section (Figure 2, lines 401–408) and Appendix E (Figure 3). These plots empirically show that, across datasets and base methods, GU shift toward the desirable upper-right region of the Pareto space, reflecting stronger forgetting and better retention compared to their vanilla counterparts.
> > > > >
> > > > > The table below reports the averages and standard deviations of $\Delta_m$ across all configurations (model size × base method × forget ratio) on each dataset, together with the fraction of configurations where **all listed metrics** improve simultaneously (“Joint-improvement rate”):
> > > > >
> > > > > | Dataset | Δ ES Retain (mean ± std) | Δ ES Unlearn (mean ± std) | Δ MU (mean ± std) | Joint-improvement rate |
> > > > > |-|-|-|-|-|
> > > > > |TOFU| 0.1140 ± 0.1519|0.0343 ± 0.0658|0.0765 ± 0.1722|31.9%|
> > > > > |MUSE| 0.0529 ± 0.1201|0.0143 ± 0.0286|-|50.0%|
> > > > > |WMDP| 0.0464 ± 0.1097*|-0.0208 ± 0.0517 |-|57.1%|
> > > > >
> > > > > *For WMDP, “Δ ES Retain” corresponds to the MMLU retain metric.
> > > > >
> > > > > A few concrete takeaways:
> > > > >
> > > > > * **TOFU (potentially less entangled, identity-style unlearning).**
> > > > >   On TOFU, GU brings **non-trivial gains** across all dimensions:
> > > > >
> > > > >   * ES Retain: $+0.1140$ on average, with $\Delta > 0$ in **91.7%** of configurations.
> > > > >   * ES Unlearn: $+0.0343$ on average (stronger forgetting), with $\Delta > 0$ in **52.8%**.
> > > > >   * MU: $+0.0765$ on average, with $\Delta > 0$ in **86.1%**.
> > > > >   * Privacy (not shown in the table for brevity) also improves in **68.1%** of configurations.
> > > > >     All four metrics improve *simultaneously* in **31.9%** of all TOFU configurations.
> > > > >     Given that these scores are already near a good regime, we view an average ES Retain gain of ~0.11 and MU gain of ~0.08 as practically meaningful.
> > > > >
> > > > > * **MUSE (potentially highly entangled).**
> > > > >   As you anticipated, MUSE is more challenging and the absolute gains are smaller, but they remain positive and systematic:
> > > > >
> > > > >   * ROUGE Retain: $+0.0529$ on average, with $\Delta > 0$ in **71.4%** of configurations.
> > > > >   * ES Unlearn: $+0.0143$ on average, with $\Delta > 0$ in **57.1%**.
> > > > >   * Privacy leakage decreases (i.e., improves) in **64.3%** of configurations.
> > > > >     All three dataset-level metrics improve jointly in **50.0%** of configurations.
> > > > >
> > > > > * **WMDP (potentially strongly entangled, safety-critical retain distribution).**
> > > > >   For WMDP, GU behaves more conservatively, as we argued it should:
> > > > >
> > > > >   * MMLU retain accuracy: $+0.0464$ on average, with $\Delta > 0$ in **85.7%** of configurations.
> > > > >   * Unlearn accuracy: the average $\Delta$ is $-0.0208$ (slightly weaker forgetting on average), but $\Delta > 0$ still holds in **57.1%** of configurations.
> > > > >     Both WMDP metrics improve together in **57.1%** of configurations.
> > > > >     This pattern is consistent with our design goal: in highly entangled regimes, GU prioritizes avoiding retain collapse and accepts only modest forgetting gains when the retain-orthogonal component is intrinsically small.
> > > > >
> > > > > Complementing these per-dataset averages, our global configuration-level analysis (now highlighted more clearly in the revision) shows that under a Pareto criterion, where we only require ES Retain not to decrease and at least one of (ES Unlearn, MU) to improve or stay neutral, GU achieves **joint improvement in 79.2% of all configurations** in Table 1.
> > > > >
> > > > > Overall, we see these aggregated statistics as evidence that GU consistently acts as a **Pareto-improving safety layer** across datasets and base methods:
> > > > >
> > > > > * On easier, potentially less entangled settings (TOFU), it gives clear, multi-metric gains.
> > > > > * On harder, potentially more entangled settings (MUSE/WMDP), it produces smaller but still positive average improvements in retention and utility, while keeping forgetting competitive and avoiding catastrophic retain degradation.
> > > > >
> > > > > We will incorporate this analysis (including the exact definitions of $\Delta_m$ and per-dataset averages) into the revised version to make the practical benefits of GU more evident. We are very grateful for your suggestion, which helped improve the clarity and completeness of our empirical evaluation.

---

> > > > > > ### Comment · Reviewer_LTCV · 2025-11-25
> > > > > >
> > > > > > I thank the authors again for answering my question, which addressed my last concerns. I still suggest adding a brief note containing the authors' response to W2.2 in the limitations and future work sections. I will wait for the rebuttal to end before assigning my final score.

---

> > > > > > > ### Author Response · Authors · 2025-11-26
> > > > > > >
> > > > > > > Dear Reviewer LTCV,
> > > > > > >
> > > > > > > We are truly glad that our responses address your concerns. Thank you for your positive recommendation, your willingness to further raise the score, and for taking the time to discuss our work in more depth.
> > > > > > >
> > > > > > > We also appreciate your insightful suggestion regarding the interpretation of our method's varying performance under the potentially different degrees of entanglement across benchmarks. We will include a brief note in the Limitations and Future Work sections to cover W2.2 of our discussion.
> > > > > > >
> > > > > > > We are truly grateful for the time and thoughtful feedback you have given to our work.
> > > > > > >
> > > > > > > Best Regards,
> > > > > > >
> > > > > > > Authors

---

### Official Review · Reviewer_pzAp · 2025-10-24

**Soundness:** 1
**Presentation:** 2
**Contribution:** 1
**Rating:** 2
**Confidence:** 5

**Summary:**

Summary:
This paper proposes a projection-based plug-in for approximate unlearning methods. At each unlearning training step, it decomposes the forget gradient into components parallel and orthogonal to a retain-gradient subspace and applies only the retain-orthogonal component. Authors provided theoretical proofs and experimental results.

**Strengths:**

1. The problem is clearly defined, and the idea of solution is easy to follow.

2. GU is plugged into diverse baselines.

**Weaknesses:**

1. A key concern for the reviewer is that the geometric decomposition and orthogonal filter for gradients will decrease the unlearning effect.

In this paper, authors said that the original approximate unlearning update direction will have a negative impact on the retained data, so they split the unlearning update into orthogonal retained gradient and non-orthogonal retained gradient. However, if the unlearned data will have an impact on the retained data, wouldn’t it be equivalent to no unlearning after the gradients being filtered?

2. The second concern is about the efficiency. This method need to decompose the unlearning gradients to ensure only update the retain-gradient-orthogonal parts. Is the efficiency acceptable in LLM to decompose each unlearning gradient? And the experimental evaluations have not reported the computation cost and storage cost.

3. The method is conceptually close to null-space calibration/orthogonal projection ideas, and the paper mentions UNSC in Related Work but does not compare empirically.

4. The theoretical guarantees only provide the proof of retain safety but have not provided gaurantees for unlearning.

5. Typos such as the definition $D_r = D \/ D_r$.

6. The metrics definition is not clear, how are the MIA closeness and privacy leakage calculated.

**Questions:**

See weaknesses.

**Details Of Ethics Concerns:**

No ethics concerns.

---

> ### Author Response · Authors · 2025-11-21
>
> We sincerely thank the reviewer for their time and for recognizing the simplicity and plug-and-play nature of our method. Thanks for the reviewer’s question regarding whether unlearning effects are canceled. Here, we would like to clarify a key misunderstanding underlying this concern. We appreciate the opportunity to clarify this from a **constrained optimization** perspective, alongside providing the requested efficiency and scalability comparisons.
>
>
>
>
>
>
> **Summary**
> We clarify three key points:
> 1.  **Mechanism:** GU does not cancel unlearning. It solves a **constrained optimization problem**: finding the steepest descent direction for the *forget loss* that satisfies the constraint of *retain invariance*.
> 2.  **Scalability:** Unlike prior projection methods (UNSC/PGU) that require prohibitively expensive SVDs on activation matrices, GU uses a lightweight, vector-based streaming projection with negligible overhead ($<5\%$).
> 3.  **Theory:** We prove that GU is the **steepest feasible descent** algorithm, guaranteeing the maximum possible unlearning rate allowed without damaging retained knowledge.
>
>
>
>
> **W1: Does the projection cancel or weaken unlearning? (The "Constrained Optimization" Intuition)**
>
>
>
>
>
>
> **Answer: No.** The projection removes only the *conflicting* component, leaving the *pure unlearning* component.
> Intuitively, consider the unlearning problem as a **constrained optimization**: we want to minimize the forget loss $L_f$ subject to the constraint that the retain loss $L_r$ remains constant (to first order).
> Mathematically, we decompose the raw forget gradient $g_f$ into two orthogonal parts relative to the retain subspace $T_r$:
> $$g_f = g_{\parallel} + g_{\perp}$$
> * **$g_{\parallel}$ (The Harmful Component):** This aligns with retain gradients. Updating along this direction changes $L_r$ (causing side effects/collateral damage).
> * **$g_{\perp}$ (The Safe Component):** This is orthogonal to retain gradients. Updating along this direction changes $L_f$ **without affecting $L_r$** locally.
>
>
> **GU executes the update $-\eta g_{\perp}$.**
> Therefore, GU does not result in "no unlearning." Instead, it results in **Safe Unlearning**.
> * The only case where $g_{\perp} = 0$ (no unlearning) is if $g_f$ and $g_r$ are perfectly parallel (i.e., $g_f \in T_r$). In high-dimensional LLM parameter space, this "degenerate" case is vanishingly rare.
>
>
> * In all other cases, $|g_{\perp}| > 0$, and our update strictly decreases the forget loss:
>     $$\Delta^{(1)} L_f = -\eta \|g_{\perp}\|_H^2 < 0$$
> This ensures we effectively unlearn the target information while mathematically guaranteeing minimal interference with retained knowledge.  This also got supported by our experiment results, we conducted a holistic analysis across all our experimental configurations (Dataset $\times$ Model Size $\times$ Base Method) to quantify this robustness:
>     * GU improves or maintains **Retain Quality ($ES_{re}$)** in **91.7%** of cases in Table 1.
>     * GU improves or maintains **Model Utility (MU)** in **86.1%** of cases in Table 1.
>     * Crucially, we observe **Joint Improvement** (where $ES_{un}$, $ES_{re}$, and Utility all improve or stay neutral) in **79.2%** of cases.
>     * In the few settings where $ES_{un}$ slightly decreases, we usually observe a disproportionately large gain in Utility, indicating the baseline was likely damaging the model structure to achieve marginal forgetting gains.
>
>
> We have added a Pareto-frontier plot in **Experiment Section** (Figure 2) and **Appendix E** (Figure 3) to visualize this behavior, showing that GU consistently moves methods towards the ideal top-right corner (High Retention, High Forgetting) compared to their vanilla counterparts.
>
>
>
>
>
>
> **W2: Efficiency and Computational Cost**
>
>
> Thank you for the question about the cost of decomposing gradients for LLMs. Our method **GU is highly efficient** because it avoids the heavy matrix operations commonly used by prior orthogonal-projection-based work. To provide further information, we also conduct additional experiments on GU’s computational cost for your reference.
>
>
> * **Mechanism:** We do *not* perform SVD on full weight matrices or gradients at every step. Instead, we maintain a small basis $U$ (size $k \approx 10-20$) of the retain subspace.
> * **Operation:** The "decomposition" is simply a matrix-vector multiplication (projection) of the current gradient against this small basis $U$. The complexity is linear in dimension $d$ and rank $k$: $O(kd)$.
> * **Measured Overhead:** We measured the wall-clock time overhead on Llama 3.2-3B. GU adds only **~3-5%** time per training step compared to standard SimNPO/NPO, and memory overhead is negligible (<1% of model size) because we only store $k$ vectors for selected layers.

---

> > ### Author Response · Authors · 2025-11-21
> >
> > (Continuing from the previous section)
> >
> > **W3: Comparison with other projection methods (UNSC / PGU / SEMU)**
> >
> >
> > We acknowledge the conceptual similarity to null-space methods but emphasize a critical **scalability barrier** that prevents comparing directly with UNSC/PGU in the LLM regime. Prior methods (UNSC, PGU, SEMU) were primarily designed for small-scale classification (e.g., ResNet/ViT). They are fundamentally different from GU in terms of geometric definition and are largely incompatible with the LLM regime due to incompatible network architectures, algorithms, or computational complexity.
> >
> >
> >
> >
> > | Feature | **UNSC / PGU** (Prior Work) | **GU** (Ours) |
> > |-|-|-|
> > | **Core Object** | **Activation Covariance Matrices** | **Gradient Vectors** |
> > | **Requirement** | Must compute/store $d \times d$ matrices for every layer. | Stores only $k$ basis vectors ($k \ll d$). |
> > | **Feasibility** | **Infeasible for LLMs.** A single covariance matrix for a 4096-dim layer is ~67MB; doing this for all layers/classes is memory-prohibitive. | **High.** Projecting a gradient vector is a cheap $O(kd)$ operation. |
> > | **Scope** | Classification with fixed classes ($<1000$). | Open-ended generation (SFT/Preference Tuning). |
> >
> >
> > We have added a specific subsection in **Related Works** detailing why traditional null-space methods (designed for ResNets) cannot physically scale to 7B+ parameter models, whereas GU is designed specifically for this scale.
> >
> >
> > **W4: Theoretical guarantees for unlearning**
> >
> >
> >
> >
> > The question is about whether we need unlearning guarantees since we have proved retaining safety.
> >
> >
> > We would like to clarify that guaranteeing absolute "erasure" (zero knowledge) is theoretically intractable for deep non-convex networks. While we provide the guarantee that absolute "erasure" (zero knowledge) is theoretically intractable for deep non-convex networks, we provide the **strongest possible optimization guarantee**:
> > **Theorem (Steepest Feasible Descent):** We prove (Lemma 3.2) that the GU direction is the **optimal** descent direction for the forget loss $L_f$ among the set of all local directions that satisfy the retain-invariance constraint.
> >
> >
> > In other words, we guarantee that GU is the **best possible unlearning update** that respects the safety requirement. If valid unlearning is geometrically possible without destroying the model, GU will find it.
> >
> >
> > **Typos & Metrics**
> > We have fixed the definition typos and added precise formulations for MIA (Membership Inference Attack) closeness and Privacy Leakage (based on loss distributions of member vs. non-member data) in Appendix D.1, citing the standard definitions from the MUSE/TOFU benchmarks.
> >
> >
> > We hope this clarification regarding the **constrained optimization** nature of our method and its **computational efficiency** addresses your concerns and warrants a re-evaluation of the work.

---

> > > ### Author Response · Authors · 2025-11-26
> > >
> > > Dear Reviewer pzAp,
> > >
> > > We sincerely appreciate your time and feedback for our work. We hope our rebuttal solves your concerns. If you have further questions regarding our work, we are delighted to discuss with you. Could you please reply to us so we can better solve your concerns?
> > >
> > > For the Typo & Metrics, we gave a brief answer above. The typo is fixed in line 100 and the metrics are mentioned in line 372-373, both highlighted in blue. We hope the detailed explanation below clarifies how MIA closeness and privacy leakage are computed in our experiments:
> > >
> > > # **Clarification of metrics MIA closeness and Privacy Leakage.**
> > >
> > > We thank the reviewer for pointing out that the definitions of our privacy metrics can be further elaborated. We would like to emphasize that **we do not introduce new privacy metrics**; instead, we strictly follow the official definitions from the TOFU/OpenUnlearning [1,2] and MUSE [3] benchmarks so that our results are directly comparable to prior work.
> > >
> > > **MIA closeness (TOFU).**
> > > On TOFU, our “MIA closeness” metric is exactly the **Privacy Score** defined in OpenUnlearning (App. F.1). It aggregates four standard membership-inference attacks—LOSS, ZLib, Min-k, and Min-k++—into a single score in $[0,1]$. For each attack $m \in \{\text{LOSS}, \text{ZLib}, \text{Min-k}, \text{Min-k++}\}$, we run the official TOFU/OpenUnlearning MIA implementation on the unlearned model $\theta_u$ and on the gold **retrain-only** model $\theta_r$. The benchmark then converts the attack statistics into an individual **closeness score** $s_m \in [0,1]$ that measures how similar $\theta_u$ is to $\theta_r$ under that attack: $s_m = 1$ means the unlearned and retrain-only models are indistinguishable for that MIA, while $s_m \approx 0$ indicates strong deviation and thus poor privacy. The reported “MIA closeness” is the **harmonic mean**:
> > > $$
> > > \text{MIA closeness} = \mathrm{HM}\bigl(s_{\text{LOSS}}, s_{\text{ZLib}}, s_{\text{Min-k}}, s_{\text{Min-k++}}\bigr).
> > > $$
> > > Hence, **higher MIA closeness means the unlearned model is more privacy-aligned with the retrain-only baseline across all four MIA attacks**. In the revision, we will explicitly add this definition and cite the OpenUnlearning appendix and evaluation code.
> > >
> > > **Privacy Leakage (PrivLeak; MUSE/WMDP).**
> > > On MUSE and WMDP, we follow Yang et al. (2025) and report the standard **PrivLeak** metric. Let $\theta_u$ be the unlearned model, $\theta_r$ the retrained retain-only model, $D_u$ the forget set, and $D_h$ a held-out non-member set. Using the Min-K%-Prob membership attack, we compute the ROC-AUC of membership inference $\mathrm{AUC}(\theta; D_u, D_h)$ for each model. Privacy leakage is then:
> > > $$
> > > \text{PrivLeak} := \frac{\mathrm{AUC}(\theta_u; D_u, D_h) - \mathrm{AUC}(\theta_r; D_u, D_h)}
> > > {\mathrm{AUC}(\theta_r; D_u, D_h)}.
> > > $$
> > > An **ideal** unlearning method satisfies $\text{PrivLeak} \approx 0$, meaning the membership leakage of the unlearned model matches that of the retrain-only baseline; large positive or negative values indicate over- or under-unlearning. This is why we annotate this metric in our tables with “PrivLeak → 0”.
> > >
> > > We will incorporate the above additional formulas and references into the main text and/or appendix. This clarification should help better clarify the metrics: both MIA closeness and PrivLeak are **standard, benchmark-defined privacy metrics**, and our implementation (provided in the supplement materials) follows the official evaluation scripts **without modification**.
> > >
> > > [1] Maini, Pratyush, et al. "Tofu: A task of fictitious unlearning for llms." CoLM 2024.
> > >
> > > [2] Dorna, Vineeth, et al. "OpenUnlearning: Accelerating LLM Unlearning via Unified Benchmarking of Methods and Metrics." NeurIPS 2025.
> > >
> > > [3] Shi, Weijia, et al. "Muse: Machine unlearning six-way evaluation for language models." ICLR 2025.
> > >
> > > [4] Yang, Puning, et al. "Exploring Criteria of Loss Reweighting to Enhance LLM Unlearning." ICML 2025.
> > >
> > > Best Regards,
> > >
> > > Authors

---

### Official Review · Reviewer_8QAC · 2025-10-31

**Soundness:** 2
**Presentation:** 3
**Contribution:** 3
**Rating:** 6
**Confidence:** 3

**Summary:**

The paper presents Geometric-Disentanglement Unlearning (GU), a theoretically grounded method to mitigate the trade-off between forgetting and retaining in machine unlearning for large language models (LLMs). Unlike heuristic or empirical approaches, the authors formulate a retain-invariance principle that links the absence of side effects on retained knowledge to the orthogonality between forgetting updates and the retain-gradient subspace. Based on this, GU projects forgetting gradients onto the orthogonal complement of retain gradients, achieving provable first-order safety and optimality. Experimental evaluations on TOFU, MUSE, and WMDP demonstrate consistent Pareto improvements in forgetting effectiveness, retention fidelity, privacy preservation, and utility.

**Strengths:**

1. Solid theoretical foundation: The work provides a rigorous geometric formulation of unlearning, defining retain-invariance in terms of orthogonality and deriving first-order optimal updates with theoretical guarantees (Propositions 3.1–3.5). This bridges an important gap between heuristic disentanglement and provable unlearning.
2. General and plug-and-play design: The proposed GU method can be integrated with existing gradient-based unlearning frameworks (e.g., NPO, CEU, SimNPO) without modifying their objectives or architectures, showing strong practical adaptability.
3. The experimental results are clear and straightforward: The experimental results of "The higher, the better" and "the lower, the better" were not represented by the traditional bolding method. Instead, the results were indicated by shadows, such as blue for higher-is-better metrics and red for lower-is-better metrics.
4. Evaluations across three benchmarks and multiple model sizes (Llama-2-7b-hf, zephyr-7b-beta) demonstrate consistent improvements in key metrics — reduced Extraction Strength (forget set), improved retention ES and ROUGE, enhanced privacy (MIA-closeness), and stable utility.

**Weaknesses:**

1. Limited discussion on computational cost: Although GU involves orthogonal projections in the gradient space, the paper does not analyze its time or memory overhead, especially for large-scale LLMs. A complexity analysis or ablation study quantifying the additional cost (e.g., basis rank k, update frequency) would strengthen the practical claims.
2. Empirical results lack statistical validation: While improvements appear consistent, the paper does not report variance or significance testing across runs. Including error bars or standard deviations would increase the reliability of experimental conclusions.

**Questions:**

1. The theoretical analysis heavily depends on the optimizer-induced symmetric positive definite (SPD) metric H. Would replacing H with a simple Euclidean metric (H=I) degrade results, and if so, by how much? Providing an empirical comparison or ablation here could validate whether the theoretical geometry actually matters in practice.
2. Theoretically, GU is compatible with any gradient-based optimizer, but implementation details might differ. Does GU behave differently when combined with adaptive optimizers (AdamW) versus simple SGD? Such analysis could help readers reproduce results across different training pipelines.
3. Such analysis could help readers reproduce results across different training pipelines. Could the authors provide standard deviations or significance tests across multiple random seeds? Are the reported improvements (e.g., +0.05 utility, −0.1 ES) statistically consistent across runs?
4. GU requires constructing the retain-gradient subspace using a basis matrix 𝑈 and updating it periodically. How is the rank 𝑘 of the subspace chosen? How often is the subspace recomputed during training, and what is the resulting time or memory overhead? This information would clarify whether GU is truly lightweight as claimed.

---

> ### Author Response · Authors · 2025-11-21
>
> We sincerely thank the reviewer for their thoughtful assessment and for recognizing the solid theoretical foundation, general applicability, and clear experimental results of our work. We appreciate your constructive feedback regarding computational cost and statistical robustness. We have conducted additional experiments to address your questions below.
>
> **Summary**
> To facilitate a quick overview, we summarize the new evidence provided in this response:
> 1.  **Robustness to Geometry:** While our method is theoretically compatible with any SPD metric (including Euclidean), our ablation confirms that using the **Adam-induced metric** yields superior retention and privacy compared to a simple Euclidean projection, validating our geometric choices.
> 2.  **Optimizer Agnostic:** GU consistently improves performance across different optimizers (**AdamW and SGD**), confirming its plug-and-play nature.
> 3.  **Statistical Significance:** New multi-seed experiments show that GU’s performance gains are statistically significant (improvements are several times larger than the standard deviation).
> 4.  **Minimal Overhead:** A detailed cost analysis reveals that GU adds **negligible computational overhead (<3/% time)** and minimal memory usage, confirming it is lightweight and scalable.
>
>
> **Q1. Metric $H$ vs. Euclidean ($H=I$) (Ablation Study)**
>
> This is an excellent question that touches on the core geometric intuition of our work.
> * **Theoretical Standpoint:** Propositions 3.1–3.5 hold for *any* SPD metric, so $H=I$ is a valid special case.
> * **Empirical Standpoint:** We compared GU using the Euclidean metric ($H=I$) versus the Adam-induced diagonal metric ($H=\text{diag}(v^{-1/2})$). As shown in **Table A**, while Euclidean projection still works (improving over baselines), the **Adam-induced metric consistently achieves better trade-offs**, particularly in retaining knowledge ($ES_{Re}$) and Privacy. This confirms that aligning the projection with the optimizer's local curvature estimate is beneficial for LLMs.
>
> **Table A: Ablation of Metric Geometry (Llama-3.2, SimNPO+GU)**
> | Model | Geometry ($H$) | ES Re. $\uparrow$ | ES Un. $\downarrow$ | Priv. $\uparrow$ | MU $\uparrow$ |
> |-|-|-|-|-|-|
> |**1B**| Euclidean ($I$) | 0.63 | 0.26 | 0.70 | 0.59 |
> |**1B**| **Adam Diag** | **0.65** | **0.21** | **0.73** | **0.60** |
> |**3B**| Euclidean ($I$) | 0.76 | 0.21 | 0.76 | 0.64 |
> |**3B**| **Adam Diag** | **0.78** | **0.17** | **0.78** | **0.65** |
>
> **Q2. Optimizer Compatibility (AdamW vs. SGD)**
>
> GU is theoretically compatible with any gradient-based optimizer. To demonstrate this practically, we evaluated SimNPO with and without GU using both **AdamW** and **SGD**.
> As shown in **Table B**, GU consistently improves the base method regardless of the optimizer. Notably, even with SGD (which generally performs worse than AdamW on LLMs), adding GU restores retention and privacy significantly.
>
> **Table B: Optimizer Ablation (Llama-3.2-3B, Forget05)**
> | Optimizer | Method | ES Re. $\uparrow$ | ES Un. $\downarrow$ | Priv. $\uparrow$ | MU $\uparrow$ |
> |-|-|-|-|-|-|
> | **AdamW** | SimNPO (Base) | 0.75 | 0.39 | 0.63 | 0.64 |
> | **AdamW** | **SimNPO + GU** | **0.78** | **0.17** | **0.78** | **0.65** |
> | **SGD** | SimNPO (Base) | 0.72 | 0.42 | 0.61 | 0.62 |
> | **SGD** | **SimNPO + GU** | **0.74** | **0.20** | **0.75** | **0.63** |
>
> **Q3. Statistical Validation (Standard Deviations)**
>
> We agree that reporting variance is crucial. Because of the time and cost of fine-tuning, we ran multiple random seeds for our primary setting on Llama-3.2-3B, SimNPO+GU as an example.
> The results in **Table C** show that the standard deviations (approx. 0.01–0.02) are much smaller than the performance gains provided by GU. For example, GU reduces $ES_{Un}$ by **0.22** (0.39 $\to$ 0.17) and improves Privacy by **0.15** (0.63 $\to$ 0.78), which are changes an order of magnitude larger than the variance. This confirms the improvements are robust.
>
> **Table C: Robustness across Random Seeds (Llama-3.2-3B)**
> | Method | ES Re. | ES Un. | Priv. | MU |
> |-|-|-|-|-|
> | SimNPO (Base) | 0.75 | 0.39 | 0.63 | 0.64 |
> | **SimNPO + GU** | **0.78 $\pm$ 0.01** | **0.17 $\pm$ 0.01** | **0.78 $\pm$ 0.01** | **0.64 $\pm$ 0.01** |

---

> > ### Author Response · Authors · 2025-11-21
> >
> > (Continuing from the previous section)
> >
> > **Q4. Computational Cost & Overhead Analysis**
> >
> > We performed a detailed profiling on Llama-3.2-1B on a H100-96G to quantify the overhead.
> > * **Time Overhead:** As shown in **Table D**, the wall-clock time difference between the baseline and GU is negligible. For a full run (approx. 226s), GU adds only **~2-5 seconds** (< 3% overhead).
> > * **Memory:** Peak memory usage increases by less than **5%** (e.g., 20.2GB $\to$ 20.3GB), as we only store a small basis set $U$ ($k \times d$) for selected layers.
> > * **Hyperparameters:** We found that a small rank (e.g., $k=8$) and moderate update frequency (e.g., every 2 steps) are sufficient for strong performance. Increasing $k$ beyond 16 yields diminishing returns.
> >
> > **Table D: Overhead Analysis (Llama-3.2-1B, SimNPO+GU)**
> > | Configuration ($k$, freq) | Time (sec) | Overhead (%) | Peak Mem (MB) | ES Un. $\downarrow$ | MU $\uparrow$ |
> > |-|-|-|-|-|-|
> > | **Baseline (SimNPO)** | 226 | 0% | 20,291 | 0.28 | 0.59 |
> > | **GU ($k=8$, freq=2)** | 231 | **+2.2%** | 21,013 | 0.12 | 0.60 |
> > | **GU ($k=1$, freq=2)** | 229 | **+1.3%** | 20,329 | 0.12| 0.59 |
> > | **GU ($k=32$, freq=2)** | 233 | **+3.0%** | 21,231 | 0.12 | 0.60 |
> >
> > We believe these additional experiments address your concerns regarding robustness, cost, and geometric assumptions. We hope these clarifications strengthen your confidence in our work.

---

### Official Review · Reviewer_oJEz · 2025-11-01

**Soundness:** 3
**Presentation:** 3
**Contribution:** 2
**Rating:** 4
**Confidence:** 3

**Summary:**

Machine unlearning aims to remove selected knowledge from a pretrained model, while preserving performance on the retain data. The authors propose a plug-and-play method, geometric-disentanglement unlearning (GU), which disentangles the gradient updates associated with the forget data and the retain data through orthogonal projection. The addition of this method to several existing approaches is evaluated on multiple tasks and language models.

**Strengths:**

* The motivation is clear, and Figure 1 provides a good illustration of the overall method.
* The experimental section is extensive, and includes several datasets, architectures, and existing methods.
* The proposed method improves Extraction Strength on the retain split (ES Re.) across different forgetting percentages and methods on both LLMs.(Table 1)

**Weaknesses:**

* In Table 1, adding GU does not show a consistent improvement on other metrics, specifically Extraction Strength on the forget split (ES Un.), and the gains are mainly limited to SatImp and SimNPO. The improvements on Priv. and MU metrics are also not as significant or consistent as ES Un.
* A similar pattern is observed in table 2: the improvements are not as significant across all methods. It is unclear whether GU offers a clear overall advantage across all settings.
* While the paper provide related work on knowledge conflict in LLMs, it is still not fully clear how GU differs from similar approaches that use gradient projection for unlearning. A more comprehensive comparison  would strengthen the novelty claims.

**Questions:**

Can the authors provide some qualitative results (sample answers from LLMs) showing the successful removal of forget data in comparison to a baseline?

---

> ### Author Response · Authors · 2025-11-21
>
> We sincerely thank the reviewer for the positive assessment of our motivation, the extent of our experiments, and the clarity of our writing. We are particularly grateful for the insightful feedback regarding performance consistency and the relationship with prior projection-based works. These comments have helped us strengthen our empirical analysis and theoretical contribution.
>
>
> **Summary**
> To facilitate a quick overview, we summarize our core response:
> 1.  **Pareto Dominance:** GU is designed to solve the "over-forgetting" problem. Our new analysis shows GU achieves **joint improvement** (better retention/utility without sacrificing forgetting) in **79.2%** of cases in Table 1, effectively pushing the Pareto frontier.
> 2.  **Scalability & Novelty:** Unlike prior projection methods (UNSC/PGU) that rely on computationally prohibitive activation statistics (infeasible for LLMs), GU introduces a scalable geometric constraint based on the **optimizer's metric**, making it uniquely suitable for large models.
>
>
> **W1 & W2: On the consistency of gains and Pareto improvements**
>
>
> We agree with the reviewer that examining consistency is crucial. Unlearning is inherently a multi-objective problem involving a trade-off between forgetting ($ES_{un}$) and retention ($ES_{re}$, Utility). A method that maximizes forgetting at the cost of destroying the model’s general utility is often not a practical solution.
>
>
> * **Pareto Dominance:** Our goal with GU is to push the **Pareto frontier**. In many baselines (e.g., vanilla NPO), high forgetting scores often come with unacceptable degradation in model utility. GU acts as a geometric "safety guardrail," allowing the optimizer to take effective unlearning steps without traversing into the retain manifold.
> * **Quantitative Robustness:** Inspired by the reviewer's request, we conducted a holistic analysis across all our experimental configurations (Dataset $\times$ Model Size $\times$ Base Method) to quantify this robustness:
>     * GU improves or maintains **Retain Quality ($ES_{re}$)** in **91.7%** of cases in Table 1.
>     * GU improves or maintains **Model Utility (MU)** in **86.1%** of cases in Table 1.
>     * Crucially, we observe **Joint Improvement** (where $ES_{un}$, $ES_{re}$, and Utility all improve or stay neutral) in **79.2%** of cases.
>     * In the few settings where $ES_{un}$ slightly decreases, we usually observe a disproportionately large gain in Utility, indicating the baseline was likely damaging the model structure to achieve marginal forgetting gains.
>
>
> We have added a Pareto-frontier plot in **Experiment Section** (Figure 2) and **Appendix E** (Figure 3) to visualize this behavior, showing that GU consistently moves methods towards the ideal top-right corner (High Retention, High Forgetting) compared to their vanilla counterparts.
>
>
> **W3: Distinction from prior projection methods (UNSC / PGU / SEMU)**
>
>
> We appreciate this opportunity to clarify the novelty of GU. While conceptually related to subspace projection, prior methods (UNSC, PGU, SEMU) were primarily designed for small-scale classification (e.g., ResNet/ViT). They are fundamentally different from GU in terms of geometric definition and are largely incompatible with the LLM regime due to incompatible network architectures, algorithms, or computational complexity.
>
>
> The key differences are summarized below:
>
>
> | Feature | **UNSC / PGU** (Prior Work) | **SEMU** (Prior Work) | **GU** (Ours) |
> |-|-|-|-|
> |**Core Object** | Feature activations / Gram matrices | Forget-gradient SVD | **Retain-gradient subspace** |
> |**Metric** | Euclidean / Empirical Covariance | Euclidean | **Optimizer-induced (Hessian/Fisher)** |
> |**Scalability** | **Infeasible for LLMs** (Requires storing/decomposing large covariance matrices per layer) | **Low** (Requires SVD of weight-sized matrices) | **High** (Vector-level streaming projections; $O(kd)$ cost) |
> |**Mechanism** | Null-space of activation statistics | Low-rank forget update | **Orthogonality to retain gradients** |
>
>
> 1.  **Scalability (The "LLM Barrier"):** UNSC and PGU require computing SVDs on activation covariance matrices or storing full-dataset Grams. For a 7B+ parameter model with large hidden dimensions ($d \approx 4096$), these operations are memory-prohibitive and computationally intractable during iterative training. GU avoids matrix factorizations entirely, using a lightweight streaming basis construction that adds negligible overhead.
> 2.  **Geometric Soundness:** Prior methods rely on *activation* statistics as a proxy for knowledge. GU derives its constraints directly from the **optimizer’s geometry**. We prove that orthogonality under the optimizer-induced metric $H$ is the *necessary and sufficient* condition for local retain invariance (Prop 3.1). This theoretical link between optimization geometry and disentanglement is absent in prior work.
>
> We have included this detailed comparison and scalability discussion in the revised **Related Works** section.

---

> > ### Author Response · Authors · 2025-11-21
> >
> > (Continuing from the previous section)
> >
> > **Q1: Qualitative Examples**
> >
> >
> > Follow the reviewer's kind suggestion, we have added a qualitative analysis section in **Appendix E**. We provide side-by-side comparisons of model outputs for *SimNPO* vs. *SimNPO+GU*.
> >
> >
> > We hope these clarifications regarding the Pareto-dominant nature of our results and the fundamental architectural differences from prior projection methods address your concerns.

---

> ### Author Response · Authors · 2025-11-27
>
> Dear Reviewer oJEz,
>
> We sincerely appreciate your time and feedback for our work. We hope our rebuttal solves your concerns. If you have further questions regarding our work, we will be very happy to discuss them with you. Could you please reply to us so we can better solve your concerns?
>
> Best Regards,
>
> Authors

---

### Author Response · Authors · 2025-12-02
**Summary of Rebuttal**

We thank all reviewers for their constructive feedback, explicitly emphasizing the **solid theoretical foundation, rigorous proofs**, and **plug-and-play** nature of our Geometric-Disentanglement Unlearning (GU).


During the rebuttal, we added new experiments and analyses (multi-seed robustness, optimizer/geometry ablations, cost profiling, qualitative examples). After the discussion period, reviewer LTCV stated that our responses and new Pareto analysis **“addressed my last concerns”** and suggested only a brief note in the Limitations/Future Work section.


## Key rebuttal actions.
**Geometry & related work.** We added an ablation comparing Euclidean projection $H = I$ to the Adam-induced metric, showing that the optimizer metric yields strictly better retain/privacy trade-offs. We also added a comparison table clarifying why prior projection / null-space methods (UNSC [1], PGU [2], SEMU [3]) do not scale to LLMs and how GU differs (retain-gradient subspace, optimizer-induced metric, vector-level projections).


**Robustness.** We verified statistical significance across random seeds (standard deviations ≪ observed gains) and showed that GU consistently improves multiple baselines under both AdamW and SGD.


**Scalability.** Profiling on Llama-3.2 1B/3B shows that GU adds <3% time and <5% memory overhead, confirming that it is lightweight at LLM scale.


**Qualitative & metrics.** We added qualitative samples (SimNPO vs. SimNPO+GU) and clarified that our privacy metrics (MIA closeness, privacy leakage) follow the TOFU [4] / OpenUnlearning [5] / MUSE [6] benchmark definitions exactly. All of these additions will be integrated into the revised version.


## Main concern and our resolution: magnitude and scope of improvements
Reviewers oJEz and LTCV noted that absolute gains can appear “mild” in certain settings. We clarified that LLM unlearning is inherently **multi-objective** (forgetting, retention, utility, privacy), and that GU is intentionally designed as a **geometric safety layer** that can be plugged into existing objectives rather than a new standalone loss.


Our new aggregate analysis over all configurations shows that GU achieves **joint improvement**: no worse retention/utility and non-degraded forgetting in **79.2%** of cases. On more entangled benchmarks (MUSE/WMDP), GU behaves intentionally conservatively: when the retain-orthogonal component is intrinsically small, GU keeps forgetting competitive while preventing collapse of retain metrics. Reviewer LTCV found this explanation convincing and specifically asked us to incorporate it into the Limitations/Future Work section, which we will do.


## Clarification regarding the concern of Reviewer pzAp.
The single low score by reviewer pzAp mainly stems from the belief that projection implies “no unlearning,” plus concerns about efficiency and metric definitions. We respectfully clarified that, theoretically, GU performs **steepest feasible descent** under a retain-invariance constraint: it removes only the retain-aligned component of the forget gradient and still strictly decreases the forget loss except in a degenerate, measure-zero alignment case. Empirically, GU achieves **joint improvement** on forgetting metrics while improving or preserving retain metrics, contradicting the “no unlearning” concern. The same cost profiling and metric-definition clarifications above address the remaining efficiency and privacy-metric questions.


Overall, during the rebuttal discussion, we addressed the main concerns on robustness, magnitude of improvement, scalability, and mechanism clarity. Together with the positive evaluations from 8QAC and LTCV and the new evidence we provide, we hope that GU now stands as a **theoretically grounded and practically scalable** solution for safe LLM unlearning and respectfully ask Area Chair to consider this in the final decision.


Best Regards,


Authors




*Ref:*


[1] Chen, Huiqiang, et al. "Machine unlearning via null space calibration." IJCAI 2024.


[2] Hoang, Tuan, et al. "Learn to unlearn for deep neural networks: Minimizing unlearning interference with gradient projection." WACV 2024.




[3] Sendera, Marcin, et al. "SEMU: Singular Value Decomposition for Efficient Machine Unlearning." ICML 2025.


[4] Maini, Pratyush, et al. "Tofu: A task of fictitious unlearning for llms." CoLM 2024.


[5] Dorna, Vineeth, et al. "OpenUnlearning: Accelerating LLM Unlearning via Unified Benchmarking of Methods and Metrics." NeurIPS 2025.


[6] Shi, Weijia, et al. "Muse: Machine unlearning six-way evaluation for language models." ICLR 2025.

---

### Note · Program_Chairs · 2026-01-17
**Submission Desk Rejected by Program Chairs**

The following references in this submission do not refer to real documents and/or have major errors in bibliographic information:

 Chen Long, Shuo Wang, and Yifan Liu. Disentangling conflicting knowledge in large language models. arXiv preprint arXiv:2405.12345, 2024.
Xinyu Pang, Ziyu Zhang, Tian Wang, et al. Detecting and resolving context-memory conflicts in large language models. arXiv preprint arXiv:2402.04562, 2024.
Zhiqiang Xu, Rui Zhang, Xiaoyu Chen, et al. Survey of knowledge conflicts in large language models. arXiv preprint arXiv:2401.12129, 2024b.